# MQFL-FHE: Multimodal Quantum Federated Learning Framework with Fully Homomorphic Encryption

## Abstract

The integration of fully homomorphic encryption (FHE) in federated learning (FL) has led to significant advances in data privacy. However, during the aggregation phase, it often results in performance degradation of the aggregated model, hindering the development of robust representational generalization. In this work, we propose a novel multimodal quantum federated learning framework that utilizes quantum computing to counteract the performance drop resulting from FHE. For the first time in FL, our framework combines a multimodal quantum mixture of experts (MQMoE) model with FHE, incorporating multimodal datasets for enriched representation and task-specific learning. Our MQMoE framework enhances performance on multimodal datasets and combined genomics and brain MRI scans, especially for underrepresented categories. Our results also demonstrate that the quantum-enhanced approach mitigates the performance degradation associated with FHE and improves classification accuracy across diverse datasets, validating the potential of quantum interventions in enhancing privacy in FL.

## 1 Introduction

Federated Learning (FL) has emerged as a powerful paradigm for collaboratively training machine learning models across decentralized data sources (McMahan et al. (2023)), allowing multiple clients to enhance model performance while preserving data privacy and complying with regulations such as the general data protection regulation (GDPR). FL alone is insufficient to fully protect data privacy, as the global model and updates shared between clients and the central server remain vulnerable to inference attacks. For instance, *membership inference attacks* use the model parameters or gradients to deduce whether a specific data sample was part of the training dataset (Shokri et al. (2017)). Similarly, *attribute inference attacks* attempt to uncover sensitive attributes of individual data samples by exploiting correlations between model updates and data features (Melis et al. (2019)). Fully homomorphic encryption (FHE) was proposed by Gentry (2009) to mitigate vulnerabilities by allowing computations on encrypted data without decryption, ensuring that sensitive information remains secure even during model training and aggregation within new gen FL. It works well against *gradient leakage attacks*, in which hackers try to recover private information by looking at the gradients that are sent back and forth during training (Phong et al. (2018)). Adding FHE to FL adds a lot of extra work to computers, causing delays and less accurate models during the aggregation phase (Zhang et al. (2023)). Consequently, while FHE protects data privacy, it can hinder the overall performance of FL systems, which is crucial for applications requiring high accuracy or real-time predictions. Despite the growing interest in multimodal FL, which aims to leverage various types of data from multiple clients, there exists a gap in research that addresses the unique challenges posed by FHE in this context (Gong et al. (2024a)). Furthermore, the application of quantum computing (QC) to enhance FL frameworks has been largely unexplored. QC has the potential to accelerate specific computations, which could alleviate performance bottlenecks introduced by FHE (Dutta et al. (2024)).

To mitigate the performance degradation that occurs during model aggregation, we propose a novel integration of QC with FHE in FL, taking into account the nature of the problem. Our method employs QC to counteract the model degradation resulting from FHE, allowing for an improved collection of model updates while protecting privacy, and for the first time, we present a novel

multimodal quantum mixture of experts (MQMoE) model within the FL framework that employs FHE. We design this architecture to handle diverse multimodal data with MoE (Yu et al. (2023a)) from various clients, enhancing representational and task-specific learning performance.

The main contributions, as illustrated in Fig. 1, are as follows:

> • Conducting a comprehensive range of single-modality experiments, from classical centralized approaches to FHE-based quantum federated learning (QFL), demonstrating how QC can mitigate model accuracy degradation caused by CKKSTensor-based FHE.
> • Developing a novel algorithmic framework for seamlessly integrating multimodal datasets with QC and FL while maintaining FHE constraints.
> • Proposing a novel MQMoE in the *FL-FHE* setup that leverages quantum layer outputs in its gating mechanism, achieving enhanced representational generalizability for task-specific learning.
> • Validating our approach through experiments in the biological domain, where we create a biological MQMoE incorporating two distinct expert types specifically designed for handling sensitive medical information (genomics and brain magnetic resonance imaging (MRI) scans).

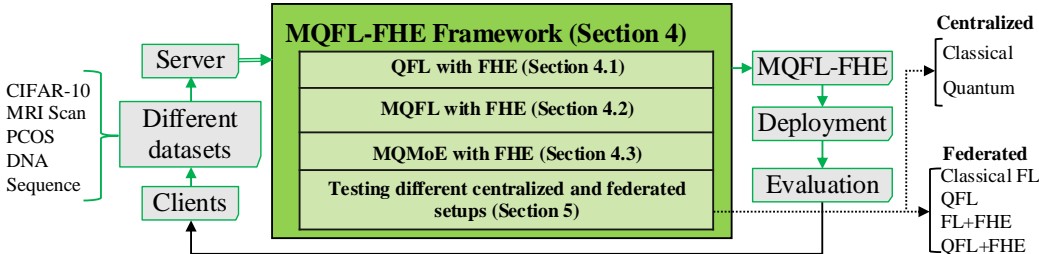

Figure 1: Overview of our novel contributions.

## 2 RELATED WORKS

**QC in FL:** Innan et al. (2024a) introduced the federated quantum neural network (FedQNN), which integrates quantum machine learning (QML) with QFL principles (Biamonte et al. (2017)). The experiments were conducted across genomics and healthcare datasets, laying the groundwork for investigations into quantum-enhanced learning methods in privacy-sensitive domains. This has been further advanced by efforts in QFL, which are creating new avenues in QML. In particular, studies like Chen & Yoo (2021); Chehimi & Saad (2022) have emphasized using QFL in quantum systems and decentralized data frameworks. Applications in healthcare and other domains, using both quantum hardware and classical simulations, have shown promising results (Lusnig et al. (2024); Bhatia & Bernal Neira (2024); Innan et al. (2024b); Javeed et al. (2024); Bhatia et al. (2023); Kais et al. (2023)). Despite this progress, challenges related to resource allocation and efficient implementation persist, as noted by Larasati et al. (2022); Ren et al. (2023); Gurung et al. (2023), highlighting ongoing efforts to address these issues.

**Privacy preservation in FL:** Rahulamathavan et al. (2023) proposed FheFL, which incorporates a multi-key additive homomorphic encryption scheme to mitigate privacy risks associated with gradient sharing in FL. The experimental results confirmed that FheFL maintains model accuracy while providing robust security against potential data breaches, emphasizing the necessity of integrating FHE techniques into federated learning frameworks to enhance data privacy. FedSHE was created by Pan et al. (2024). It is a useful FL scheme that uses adaptive segmented CKKS homomorphic encryption to keep gradients safe from leakage attacks. Their analysis of security parameters underscores the trade-offs between correctness, efficiency, and security in FL environments.

**Multimodal data handling in FL:** Yu et al. (2023b) developed CreamFL, a framework designed for multimodal FL. This method enables clients with heterogeneous model architectures to contribute to a unified global model without sharing their private data. By implementing a contrastive

representation ensemble strategy, CreamFL effectively addresses the challenges posed by modality discrepancies and task diversities. Gong et al. (2024b) further advanced this area by introducing a multimodal vertical FL framework utilizing bivariate Taylor series expansion for encryption, thus eliminating the reliance on third-party encryption providers.

Unlike the aspects discussed in previous literature, our methodology tackles the problem of managing multimodal data while reducing performance degradation due to FHE. We propose a seamless algorithm that integrates QC techniques with multimodal FL setups within the FHE domain.

## 3 PROBLEM FORMULATION

Consider a FL setup with $K$ clients, each with local data $\mathcal{D}_k$, where $k \in \{1, 2, \ldots, K\}$. Each client trains a local model with parameters $\mathbf{w}_k^t$ and shares the encrypted updates $\mathcal{E}(\mathbf{w}_k^t)$ with a central server. During the aggregation step, the global update is computed as $\mathcal{E}(\mathbf{w}^{t+1}) = \frac{1}{K} \sum_{k=1}^{K} \mathcal{E}(\mathbf{w}_k^t)$, ensuring data privacy by performing operations on encrypted weights without decryption.

FHE schemes, like Cheon-Kim-Kim-Song (CKKS), allow both homomorphic addition and multiplication with the help of cyclotomic polynomial ring topologies. For homomorphic addition, the server computes the encrypted sum $\mathcal{E}(\mathbf{w}_1 + \mathbf{w}_2) = \mathcal{E}(\mathbf{w}_1) + \mathcal{E}(\mathbf{w}_2)$. The decryption at the client side yields the sum $\mathbf{w}_1 + \mathbf{w}_2 \approx \text{Dec}(\mathcal{E}(\mathbf{w}_1) + \mathcal{E}(\mathbf{w}_2))$. For homomorphic multiplication, the encrypted product of two ciphertexts $\mathcal{E}(\mathbf{w}_1 \cdot \mathbf{w}_2)$ expands into three components. To control ciphertext growth, a re-linearization step reduces the result to two components, i.e., $\text{Relin}(\mathcal{E}(\mathbf{w}_1) \cdot \mathcal{E}(\mathbf{w}_2))$.

Despite the privacy benefits, the use of FHE leads to significant challenges like computational overhead and model accuracy. The encryption and decryption processes, along with operations on encrypted data, introduce noise and quantization errors which accumulate during operations like multiplications, resulting in the degradation of the decrypted model's accuracy $\mathbf{w}^{t+1} \approx \mathbf{w}_1 + \mathbf{w}_2 + \cdots + \mathbf{w}_K$. Therefore, the primary challenge is to mitigate this accuracy degradation, especially when multiple rounds of homomorphic operations are performed during model aggregation in FL. To address this, we propose leveraging QC to reduce noise accumulation, thus preserving model accuracy while maintaining the privacy guarantees of FHE.

## 4 METHODOLOGY

We propose a novel framework that integrates FHE with QFL to improve privacy-preserving machine learning without compromising performance. Our approach leverages quantum computations to counterbalance the degradation introduced by FHE during model aggregation, particularly in multimodal settings. The key components of the proposed methodology include the integration of FHE with QFL, multimodal quantum federated learning (MQFL), and the novel multimodal quantum mixture of experts (MQMoE).

### 4.1 QFL-FHE: INTEGRATING FULLY HOMOMORPHIC ENCRYPTION WITH QUANTUM FEDERATED LEARNING

In the QFL framework, the integration of FHE allows us to homomorphically aggregate encrypted weights across distributed clients securely, presenting a complex but highly secure approach for data privacy preservation as shown in Fig. 2 considering there to be a single type of input modality.

Let's look at the CKKS (Cheon-Kim-Kim-Song) encryption scheme. This scheme is particularly well-suited for federated learning because it allows approximate arithmetic on encrypted data, enabling repeated additions and multiplications while maintaining fidelity. Each client $i$ holds local data $x_i$ and computes model updates through a local update function $g_i$. The encrypted model updates can be expressed as:

$$Enc_{\text{CKKS}}(g_i(\theta, x_i)) = \sum_{j=0}^{N} c_j \cdot X^j + E,$$

where $c_j$ are the coefficients of the cyclotomic polynomial representation, $N$ is the degree of the polynomial, $X$ is a complex root of unity associated with the cyclotomic polynomial tied to the

Figure 2: Overview of the MQFL-FHE framework. Each client (e.g., client 1, client 2, etc.) trains a local model on its private dataset, encrypts the model weights using the CKKS homomorphic encryption scheme, and sends the encrypted local model $\mathbf{w}_k^{enc}$ to the central server. The global server securely aggregates the encrypted local models using a weighted sum based on client data contributions. The aggregated model is then decrypted, optimized, and distributed back to all clients as the updated global model $\mathbf{w}_g$. A single key setup is used for both encryption and decryption, ensuring secure communication throughout the process.

encryption parameters, and $E$ is the error term introduced by the CKKS encoding. This ensures that each client securely computes and sends back the encrypted results $Enc_{\text{CKKS}}(g_i(\theta, x_i))$ to the central server, maintaining privacy throughout the process. Upon receiving the encrypted updates from all clients, the server performs aggregation of these encrypted model updates. Utilizing the homomorphic properties of the CKKS scheme, the aggregated update can be represented as:

$$Enc_{\text{CKKS}}\left(\sum_{i=1}^{M} g_i(\theta, x_i)\right) = \sum_{j=0}^{N} c_j \cdot \left(\sum_{i=1}^{M} g_i(\theta, x_i)\right)^j + E',$$

where $E'$ is the error introduced during the aggregation process. This capability allows the server to perform necessary computations without needing to decrypt the data. The CKKS scheme is configured with a polynomial modulus degree of 8192, which defines the ring size $\mathbb{Z}[X]/(X^n + 1)$ with $n = 8192$, providing a security level of 128 bits. The coefficient modulus is split into four primes with bit sizes [60, 40, 40, 60], resulting in a total modulus size of 200 bits, balancing security and computational efficiency. The global scaling factor of $2^{40}$ ensures sufficient precision for fixed-point arithmetic. Each client further incorporates quantum principles, beginning by encoding its classical data into a quantum state $|x_i\rangle$ using a unitary operation defined as:

$$|x_i\rangle \equiv U^\dagger(x_i)|0\rangle = \bigotimes_{j=1}^{dh} R_x^\dagger(x_{ij})|0\rangle,$$

where $R_x(x_{ij})$ denotes the rotation operation applied to the input features $x_{ij}$. The quantum model update $q_i$ is then encrypted using a hybrid lattice-based encryption scheme, resulting in ciphertexts that ensure both classical and quantum model privacy, denoted as $c_i = \text{Enc}_{\text{CKKS}}(|x_i\rangle)$. For client $i$, let the encrypted model parameters be $C_i = \text{Enc}_{\text{CKKS}}(\boldsymbol{\theta}_i)$, where $\boldsymbol{\theta}_i$ denotes the local parameters of the quantum model. By applying the FedAvg aggregation algorithm to homomorphically aggregate encrypted model parameters from all participating clients, the global aggregation of these parameters is expressed as:

$$C_{\text{global}} = \text{Enc}_{\text{CKKS}}\left(\frac{1}{N}\sum_{i=1}^{N} M(\boldsymbol{\theta})\right) = \frac{1}{N}\sum_{i=1}^{N} C_i.$$

The operation of $M(\boldsymbol{\theta})$ can be expressed in terms of individual quantum gates applied to the state $|x\rangle$. This can be formalized as:

$$|y\rangle = M(\boldsymbol{\theta})|x\rangle = U_k(\theta_k)\cdots U_1(\theta_1)|x\rangle,$$

where $U_j(\theta_j)$ represents the $j$-th quantum gate acting on the state. Following the homomorphic encryption properties, the output state $|y\rangle$ is encrypted as $c_y = \text{Enc}_{\text{CKKS}}(|y\rangle)$. This capability allows for the evaluation of functions of the quantum state without direct access to the raw data. The parameterized quantum circuit (PQC) employed in the experiments is defined as:

$$M(\theta) = \prod_{l=1}^{L} \left( \prod_{i=1}^{n} R_X(\theta_{l,i}) \right) \left( \prod_{i=1}^{n-1} \text{CNOT}(i, i+1) \right),$$

where $L$ is the number of layers, $R_X(\theta)$ is the X-rotation gate, and $\text{CNOT}(i, j)$ is the controlled-NOT gate with qubit $i$ as control and qubit $j$ as target. This aggregation process effectively maintains the privacy of individual client models while enabling the creation of a global model $\boldsymbol{\theta}_{\text{global}}$.

## 4.2 MQFL-FHE: Multimodal Quantum Federated Learning with Fully Homomorphic Encryption

Building upon the *QFL-FHE* foundation, we introduce MQFL, which handles heterogeneous data modalities such as image & text data across clients, leveraging QC to enhance both representational learning and model aggregation. The integration is detailed in Algorithm 1 and illustrated in Fig. 2.

---

**Algorithm 1** Multimodal Quantum Federated Learning with Fully Homomorphic Encryption

---

1: **Require:**
2:     $ctx$: Fully homomorphic encryption context
3:     $N$: Number of federated clients
4:     $params$: Encryption parameters
5:     $G$: Quantum gate set
6:     $D$: Parameterized Quantum Circuit (PQC) depth
7:     $M$: Number of modalities (e.g., images, text)
**Ensure:** Aggregated global model $\mathbf{w}_g$
8: **Initialization:**
9: Generate CKKS context $ctx \leftarrow \text{CKKSContext}(params)$
10: Generate Galois keys for rotations keys $\leftarrow ctx.\text{generate\_galois\_keys}()$
11: Initialize global Multimodal QNN model $\mathbf{w}_g \leftarrow \text{InitializeMQNN}(D, G, M)$
12: **Client-Side QNN Training and Encryption:**
13: **for** each client $k \in \{1, \ldots, N\}$ **in parallel do**
14:     Data Preprocessing for multimodal dataset $\mathcal{D}_k \leftarrow \text{PrepareMultimodalDataset}(k)$         ▷ Handle multiple data types
15:     Train local Multimodal QNN $\mathbf{w}_k \leftarrow \text{TrainMultimodalQNN}(\mathcal{D}_k, \mathbf{w}_g, D, G)$
16:     Quantize and encrypt the local model $\mathbf{w}_k^{enc} \leftarrow \text{Encrypt}(\text{Quantize}(\mathbf{w}_k), ctx)$
17:     Send encrypted model $\mathbf{w}_k^{enc}$ to the server
18: **end for**
19: **Server-Side Aggregation:**
20: Initialize $S \leftarrow 0$                                                                 ▷ Accumulator for weighted sum
21: $n_{\text{total}} \leftarrow \sum_{k=1}^{N} n_k$                                           ▷ Total number of samples across all clients
22: **for** each client $k \in \{1, \ldots, N\}$ **do**
23:     Receive $\mathbf{w}_k^{enc}$ from client $k$
24:     Aggregate encrypted weights $S \leftarrow S + \mathbf{w}_k^{enc} \cdot \frac{n_k}{n_{\text{total}}}$
25: **end for**
26: **Client-Side Decryption and Global Model Update:**
27: **for** each client $k \in \{1, \ldots, N\}$ **in parallel do**
28:     Decrypt aggregated model $\mathbf{w}_g \leftarrow \text{Decrypt}(S, \text{secret\_key})$
29:     Update global multimodal QNN model $\mathbf{w}_g$ on the client
30: **end for**
31: **PQC Update:**
32: Adjust PQC parameters and architecture $\mathbf{w}_g \leftarrow \text{OptimizePQC}(\mathbf{w}_g, D, G)$
33: **Model Distribution:**
34: **for** each client $k \in \{1, \ldots, N\}$ **in parallel do**
35:     Send global model $\mathbf{w}_g$ to client $k$
36: **end for**
37: **Repeat** from step 11 until maximum communication rounds
38: **return** $\mathbf{w}_g$

---

In *MQFL-FHE*, we employ an immediate feature-level fusion by utilizing multi-head attention mechanisms (Vaswani (2017)) to extract relevant features from diverse input modalities. The client-specific preprocessing function is represented as $\mathcal{D}_k \leftarrow \text{PrepareMultimodalDataset}(k)$, ensuring that varying input types are effectively managed and allowing local models to learn from different sources. The local model update for client $k$ can be expressed as:

$$\mathbf{w}_k = M_k(\mathbf{w}_g, \mathcal{D}_k) = M_k \left( \mathbf{w}_g, \bigcup_{m=1}^{M} \mathcal{D}_{k,m} \right), \quad \text{Attention}(Q, K, V) = \text{softmax}\left( \frac{QK^T}{\sqrt{d_k}} \right) V,$$

where $X$ represents the input features obtained after concatenating features from multiple modalities, and $Q = XW_Q$, $K = XW_K$, and $V = XW_V$ are the query, key, and value matrices computed as learned linear projections of $X$. Here, $W_Q$, $W_K$, and $W_V$ are the projection weight matrices, and $M_k$ represents the multimodal training function that leverages extracted features from multiple modalities $\mathcal{D}_{k,m}$ such as images and text through attention-based mechanisms.

In contrast to *QFL-FHE*, where each client handles a singular data modality, *MQFL-FHE*'s architecture allows for a plug-and-play approach. This scalability is further supported by the abstraction of the **PrepareMultimodalDataset** function, which can be generalized for various data types without necessitating extensive modifications to the overall algorithm. In this paper, we define the process of handling both modalities where DNA sequences are split into k-mers, converted into text, and represented using TF-IDF vectorization, while MRI images are preprocessed using standard transformations. These modalities are then paired and prepared for training, after which the data is passed through a hybrid classical-quantum architecture. The algorithm's structure, particularly in the server-side aggregation step, can be mathematically articulated as:

$$S = \sum_{k=1}^{N} \left( \mathbf{w}_k^{enc} \cdot \frac{n_k}{n_{\text{total}}} \right) = \sum_{k=1}^{N} \left( \text{Enc}_{\text{CKKS}} \left( M_k(\boldsymbol{\theta}, \mathcal{D}_k) \right) \cdot \frac{n_k}{n_{\text{total}}} \right),$$

where $n_k$ represents the number of samples for client $k$ and $n_{\text{total}}$ is the aggregate sample size across all clients.

## 4.3 MQMoE-FL-FHE: Multimodal Quantum Mixture of Experts based Federated Learning with Fully Homomorphic Encryption

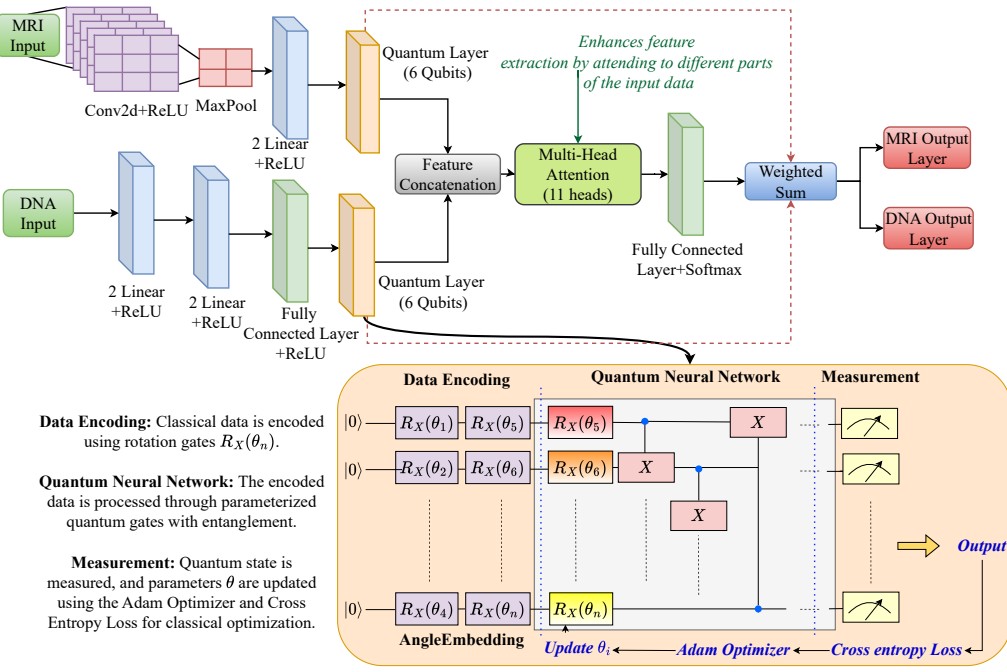

Figure 3: The model workflow follows a MQMoE approach. MRI and DNA data are processed through distinct classical layers, where MRI inputs pass through convolutional and pooling layers, while DNA inputs are processed through linear layers. After classical preprocessing, both inputs are passed through quantum layers outlined in the figure, representing QNNs, with 6-qubit quantum layers for both MRI & DNA to create two quantum expert representational vectors. These quantum layers serve as specialized experts to extract complex feature representations. The outputs from the quantum experts are combined through feature concatenation, enhanced by a multi-head attention (MHA) mechanism to capture key features from the input data. For experts, the MHA+FC layer+Softmax is the gating network. Combined with the weighted sum, this integrates the quantum expert outputs, leading to their respective output layers. This process optimizes the final predictions for both MRI and DNA output layers.

We extend the idea of *MQFL-FHE* to introduce the MQMoE framework. This approach enhances the capabilities of the previous work by employing a mixture-of-experts strategy within the FL setting while maintaining FHE for privacy preservation. In the framework of *MQMoE-FL-FHE*, the network consists of a gating mechanism to dynamically select expert models tailored to the various modalities of input data, which allows for improved representational learning. Each client $k$ maintains a set of local experts $E_k = \{e_{k,1}, e_{k,2}, \ldots, e_{k,m}\}$ designed to specialize in different modalities as shown in Fig. 3. The gating function $g_k$ and the overall output $\mathbf{y}_k$ are computed as follows:

$$g_k(x) = \text{softmax}(W_g \cdot \phi(x)) \quad \text{and} \quad \mathbf{y}_k = \sum_{j=1}^{m} g_{k,j} \cdot e_{k,j}(x),$$

where $W_g$ represents the learned weights of the gating network, $\phi(x)$ is the feature representation of the input data $x$, and $g_{k,j}$ denotes the gating weight for expert $e_{k,j}$. In contrast to the basic *MQFL-FHE* architecture, which aggregates encrypted model updates without distinguishing between modalities, *MQMoE-FL-FHE* incorporates a gating strategy that adjusts based on the specific characteristics of the incoming data. Each expert's output $e_{k,j}(x)$ is generated through a PQC defined as:

$$e_{k,j}(x) = U_j(\theta_j)|x\rangle,$$

where $U_j(\theta_j)$ denotes the quantum operation applied to the input state, allowing for encoding classical data into quantum states.

## 5 DISCUSSION

In the experimental setting of our study, we use datasets that vary significantly in terms of image or sequence count and imbalance factors to evaluate our models under a variety of conditions. The CIFAR-10 dataset, with a balanced distribution (imbalance factor of 1), includes $60,000$ images and employs a batch size of 128 for training, as shown in Table 1. In contrast, the DNA Sequence dataset (Singh (2023)) presents a higher challenge with an imbalance factor of $5.595$ and a total of $4,380$ sequences, trained with a smaller batch size of 16, to possibly mitigate the effects of its substantial class imbalance. Other datasets, such as MRI Scan and PCOS (Hub et al. (2024)) with imbalance factors of $1.207$ and $2.544$, respectively, and total counts of $7,022$ and $3,200$, use a batch size of 32. Regarding the hyperparameters, a learning rate of $1 \times 10^{-3}$ is applied, which may increase to $3 \times 10^{-3}$ if learning plateaus, as detailed in Table 2. The FL utilizes the FLWR library (Beutel et al. (2020)) to manage training across 10 clients over 20 communication rounds with 10 epochs per client, contrasting with the 25 total epochs in the centralized approach. Simultaneously, Pennylane (Bergholm et al. (2018)) is employed to execute quantum computations within the same FL setup. The experiments are conducted on hardware with an AMD EPYC 7F72 (96) @ 3.685GHz CPU and NVIDIA A100-PCIE-40GB GPUs to ensure efficient processing and computation. This setup, incorporating diverse batch sizes and a range of hyperparameters, ensures comprehensive evaluation across both federated and centralized learning settings.

Table 1: Dataset overview & batch size chosen for the datasets for every experiment.

| Dataset | Imbalance Factor (IF) | Total Images/ Sequences | Batch Size | Classes |
|---------|----------------------|------------------------|-----------|---------|
| CIFAR-10 | 1 | $60,000$ | 128 | 10 |
| MRI Scan | $1.207$ | $7,022$ | 32 | 4 |
| PCOS | $2.544$ | $3,200$ | 32 | 2 |
| DNA Sequence | $5.595$ | $4,380$ | 16 | 7 |

Table 2: Hyperparameters and settings.

| Parameter | Value |
|-----------|-------|
| Learning Rate | $1e-3$ (higher if plateauing: $3e-3$) |
| Train-Val Split | FL: $90-10$, Centralized: $80-20$ |
| Test Set | Same for both |
| Number of Clients | 10 (FL only) |
| Communication Rounds | 20 (FL only) |
| Epochs per Client | 10 (FL only) |
| Total Epochs | 25 (Centralized only) |
| Client Resources | {"num_gpu": 1 } |

The test results show that the classical and quantum models perform very differently in both centralized and federated setups. This is because different computational approaches and datasets have their unique dynamics. In the centralized experiments, as outlined in Table 1, the classical setup consistently achieves high accuracy, with CIFAR-10 reaching a test accuracy of $76.59\%$, compared to the quantum setup's $74.33\%$. The MRI Scan dataset presents the highest accuracy in the classical centralized setup, with training and testing accuracy of $99.74\%$ and $93.45\%$, respectively. In the multimodal setup combining DNA and MRI data, the classical approach reports test accuracy of $84.64\%$ for DNA and $90.62\%$ for MRI, slightly decreasing in the quantum centralized setting. In the

Table 3: Centralized experiment results for different setups and datasets.

| Experiment Setup | Dataset | Train Loss | Test Loss | Train Accuracy | Test Accuracy | Time (sec) |
|---|---|---|---|---|---|---|
| Classical Centralized | CIFAR-10 | 0.29 | 0.659 | 89.70% | 76.59% | 214.58 ± 0.03 |
| | DNA Sequence | 0.0102 | 0.508 | 99.71% | 94.50% | 97.51 ± 0.04 |
| | MRI Scan | 0.0068 | 0.566 | 99.74% | 93.45% | 412.23 ± 0.06 |
| | PCOS | 0.1263 | 0.91 | 94.79% | 74.22% | 158.17 ± 0.05 |
| | DNA+MRI Multimodal | DNA: 0.027 MRI: 0.012 | DNA: 0.855 MRI: 0.751 | DNA: 99.24% MRI: 99.71% | DNA: 84.64% MRI: 90.62% | 372.23 ± 0.07 |
| Quantum Centralized | CIFAR-10 | 0.37 | 0.707 | 87.63% | 74.33% | 850.93 ± 0.13 |
| | DNA Sequence | 1.266 | 1.33 | 94.54% | 88.63% | 524.98 ± 0.08 |
| | MRI Scan | 0.0073 | 0.2903 | 99.86% | 92.12% | 608.23 ± 0.09 |
| | PCOS | 0.47 | 1.02 | 89.66% | 72.33% | 253.38 ± 0.02 |
| | DNA+MRI Multimodal | DNA: 0.067 MRI: 0.058 | DNA: 0.979 MRI: 0.560 | DNA: 98.70% MRI: 98.54% | DNA: 86.53% MRI: 85.37% | 649.64 ± 0.12 |

FL experiments, as described in Table 2, the QFL with CIFAR-10 shows a test accuracy of 72.16%, marginally outperforming its classical counterpart at 71.03%. This indicates the quantum model's potential in federated settings despite complex data privacy constraints. For the multimodal datasets in federated settings, the DNA and MRI datasets maintain high training accuracy, with slight reductions in test accuracy under quantum models, emphasizing the nuanced challenges in applying quantum computations to complex data types. Furthermore, the experiments highlight the extended computation times associated with quantum models and encryption methods. Specifically, the QFL setup for CIFAR-10 necessitates approximately 9091.34 ± 2.11 seconds, significantly more than the classical FL's 3405.67 ± 1.89 seconds. Introducing FHE in federated setups notably increases computational demands, with QFL + FHE requiring up to 9747.32 ± 2.23 seconds for CIFAR-10. In our study, we extensively analyze the performance of federated learning models applied to DNA

Table 4: Federated experiment results for various setups and datasets.

| Experiment Setup | Dataset | Test Loss | Train Accuracy | Test Accuracy | Time (sec) |
|---|---|---|---|---|---|
| Classical FL | CIFAR-10 | 1.257 | 100.00% | 71.03% | 3405.67 ± 1.89 |
| | DNA Sequence | 1.203 | 100.00% | 94.09% | 3926.97 ± 2.42 |
| | MRI Scan | 1.524 | 100.00% | 93.75% | 5510.97 ± 2.92 |
| | PCOS | 1.416 | 100.00% | 65.37% | 2167.52 ± 1.21 |
| | DNA+MRI Multimodal | DNA: 0.416 MRI: 1.072 | DNA: 99.01% MRI: 98.75% | DNA: 94.75% MRI: 85.56% | 5563.82 ± 4.31 |
| QFL | CIFAR-10 | 1.202 | 97.15% | 72.16% | 9091.34 ± 2.11 |
| | DNA Sequence | 1.228 | 100.00% | 93.76% | 6809.63 ± 2.74 |
| | MRI Scan | 0.338 | 100.00% | 89.71% | 7215.89 ± 3.22 |
| | PCOS | 0.903 | 100.00% | 73.29% | 3433.54 ± 1.56 |
| | DNA+MRI Multimodal | DNA: 0.349 MRI: 0.906 | DNA: 99.23% MRI: 98.70% | DNA: 94.24% MRI: 85.83% | 8531.73 ± 5.34 |
| FL + FHE | CIFAR-10 | 1.322 | 97.69% | 68.53% | 4021.75 ± 1.97 |
| | DNA Sequence | 1.434 | 100.00% | 91.87% | 4421.45 ± 2.52 |
| | MRI Scan | 0.402 | 100.00% | 88.40% | 5904.54 ± 3.10 |
| | PCOS | 1.379 | 100.00% | 64.11% | 2645.88 ± 1.49 |
| | DNA+MRI Multimodal | DNA: 0.408 MRI: 1.738 | DNA: 98.75% MRI: 98.21% | DNA: 93.75% MRI: 83.33% | 7520.98 ± 4.95 |
| QFL + FHE | CIFAR-10 | 0.0937 | 97.90% | 71.12% | 9747.32 ± 2.23 |
| | DNA Sequence | 0.782 | 100.00% | 94.32% | 7123.91 ± 2.91 |
| | MRI Scan | 0.360 | 100.00% | 88.75% | 7851.86 ± 3.54 |
| | PCOS | 1.090 | 100.00% | 70.15% | 3942.60 ± 1.65 |
| | DNA+MRI Multimodal | DNA: 0.174 MRI: 0.713 | DNA: 99.64% MRI: 100% | DNA: 95.31% MRI: 87.26% | 10314.34 ± 6.28 |

and MRI datasets, with and without the integration of quantum computing enhancements and FHE. Fig.4 presents these models' receiver operating characteristic (ROC) curves, offering a detailed view of their discriminative abilities under different configurations. For the DNA dataset, the ROC analysis reveals that the classical FL-FHE model achieves a micro-average and macro-average area under the curve (AUC) of 0.92, indicating robust predictive performance across multiple classes. However, when quantum enhancements are applied through the QFL-FHE setup, there is a noticeable improvement, with the micro-average AUC increasing to 0.94 and the macro-average rising to 0.95. This enhancement validates that integrating QC can refine the model's sensitivity and specificity,

particularly in managing complex patterns within the DNA sequences. In contrast, the MRI dataset exhibits different dynamics. Under the FL-FHE setup, the micro-average and macro-average ROC AUCs are reported at 0.93, demonstrating high efficacy in class discrimination. In the QFL-FHE setup, these metrics improve to 0.97 for both micro-average and macro-average AUCs. This significant increase underscores the potential of quantum enhancements in effectively handling the intricacies of MRI data, potentially due to the quantum model's capability to process high-dimensional data more efficiently.

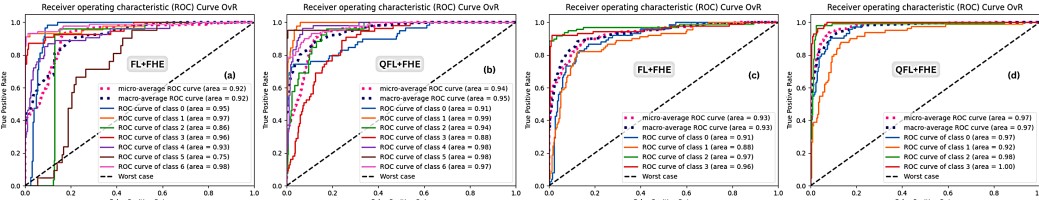

Figure 4: ROC curves for DNA and MRI datasets under FL with and without quantum enhancements. Panels (a) and (b) show the ROC curves for the DNA dataset under FL-FHE and QFL-FHE setups, respectively, highlighting the performance across various classes. Panels (c) and (d) show the ROC curves for the MRI dataset under FL-FHE and QFL-FHE setups, respectively. Each panel demonstrates the true positive rate against the false positive rate for each class, including micro-average and macro-average ROC curves, illustrating the models' discriminative ability in a privacy-preserving federated learning context.

## 5.1 ABLATION STUDY

Our analysis is crucial for understanding how each technology influences performance and privacy when diverse data types are integrated within federated and centralized settings:

- **Impact of technology removal:** Removing QC from the multimodal setup significantly reduces the system's ability to effectively integrate and process heterogeneous data in federated and centralized environments. We observe a noticeable decrease in AUC metrics, highlighting QC's role in managing complex data interactions within the framework.

- **Effects of FHE modifications:** Altering the complexity of the encryption method impacts the setting's capacity to secure data across various modalities without compromising computational efficiency in both settings. This change leads to a slight decrease in performance metrics, emphasizing the need for a robust encryption method that balances security with operational efficiency.

- **Combined effects in multimodal settings:** The simultaneous removal of QC and FHE substantially degrades the framework's functionality, particularly affecting its ability to perform secure and efficient multimodal data integration in both federated and centralized systems. This underlines the synergistic importance of both technologies in enhancing data processing and security.

- **Efficacy of the *MQMoE* approach:** The *MQMoE* approach, specifically designed for multimodal data, exhibits resilience and enhanced capability when equipped with QC and FHE. It maintains high performance in integrating and processing multimodal data, ensuring accuracy and privacy across both federated and centralized configurations. The removal or modification of these technologies from this setup elucidates their critical roles in supporting advanced multimodal FL and centralized systems.

- **Centralized systems outperform federated ones, but trend varies with QC:** Notably, analysis reveals that centralized systems generally outperform federated ones in terms of computational efficiency and ease of implementing privacy-preserving techniques. However, federated systems using QML can surpass centralized models under certain conditions. Specifically, with smaller datasets, QML in federated learning performs better due to its ability to generate richer representations, which is particularly effective for smaller problems. In federated settings, where data is split across clients, QML's strengths in handling sparse and fragmented data lead to improved performance for decentralized problems, making it more effective than centralized models for such datasets.

- *FL+FHE* vs *QFL+FHE*: To evaluate the impact of each module on the final performance of the global model, we conduct experiments on DNA+MRI multimodal datasets for an ablation study. As Fig. 5 shows, the confusion matrices show that QFL+FHE significantly improves classification accuracy compared to FL+FHE. For example, in the DNA dataset, QFL+FHE achieves better diagonal dominance, especially in class 6 with 0.31 accuracy, compared to 0.34 for FL+FHE. In the MRI dataset, QFL+FHE shows superior performance for crucial classes such as glioma and pituitary, with values of 0.26 and 0.2, respectively, compared to 0.17 and 0.03 for FL+FHE. This highlights that QFL+FHE which is FL+FHE when integrated with QC offers improved classification performance and better separation of challenging classes, particularly for more difficult or underrepresented categories.

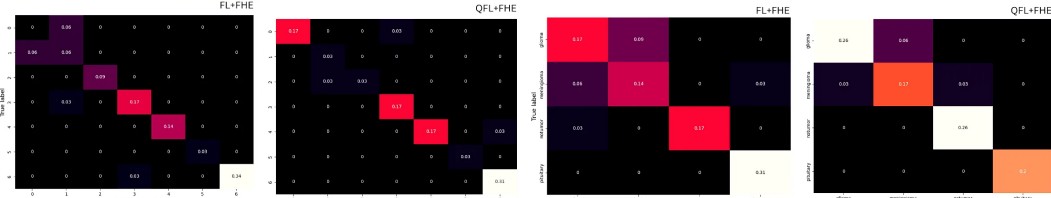

Figure 5: Confusion matrices for DNA and MRI datasets under FL with and without quantum enhancements. The first two matrices on the left illustrate the classification performance for the DNA dataset using FL+FHE and QFL+FHE settings, respectively. These matrices show the distribution of true labels versus predicted labels, with the intensity of the colors indicating the proportion of predictions. The right two matrices focus on the MRI dataset under the same setting, FL+FHE, and QFL+FHE. Each matrix highlights how effectively each model categorizes different tumor types, such as glioma, meningioma, notumor, and pituitary, with brighter colors representing higher frequencies of predictions.

This study clarifies the substantial contributions of QC and homomorphic encryption in managing multimodal datasets within both federated and centralized settings. Their integration boosts performance metrics and upholds privacy standards, making them indispensable for future advancements in secure and effective learning solutions. This is particularly evident in centralized systems, where integrating these technologies has demonstrably enhanced concurrent performance and privacy.

## 6 CONCLUSION

Our study has shown that FL, despite its advantages for collaborative model training across decentralized data sources, faces significant challenges in data privacy and computational efficiency. These challenges are particularly critical when high accuracy and quick processing are required, such as in real-time applications. The introduction of FHE has proven effective in enhancing data privacy by allowing computations on encrypted data. However, FHE increases the computational load significantly, which can reduce the performance and speed of model training. To tackle these issues, our research introduces an integration of QC with FL and FHE. This new approach has shown potential in reducing FHE's performance drawbacks and efficiently managing complex, multimodal datasets. Our MQMoE framework demonstrates how QC can improve the learning performance and privacy of federated learning systems. The results from various experimental setups, including classical and quantum models, indicate that while quantum solutions add complexity, they offer potential computational advantages. These findings suggest that quantum-enhanced federated learning could be a viable solution for handling large-scale, privacy-sensitive applications. In summary, this research points out the limitations of current FL systems and proposes a novel solution that uses QC to enhance security and efficiency. While QC is not without its challenges, it represents a promising start in the search for more effective FL technologies. This could be particularly valuable in fields where privacy and quick data processing are essential, marking a significant step forward in developing advanced FL technologies.

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

# 7 APPENDIX

## 7.1 MATHEMATICAL ANALYSIS OF ERROR PROPAGATION AND NOISE STABILIZATION IN FHE-QUANTUM HYBRID MODELS

Noise in FHE-based hybrid architectures arises primarily from errors introduced during the encryption and decryption processes of FHE. These errors accumulate as computations progress, and their propagation can significantly impact the fidelity of the results. Mathematically, such errors can be modeled as deviations in state evolution during computation, represented by transformations in the Hilbert space. However, the inherent properties of quantum operations, governed by the special unitary group $SU(2)$, impose strict constraints on the behavior of noise. Quantum operations on qubits correspond to rotations on the Bloch sphere, represented by the unitary matrix $U(\phi, \theta, \psi)$(Sultanow et al. (2024)). These operations adhere to the condition $U^\dagger U = I$, ensuring the preservation of the norm of the quantum state vector. Specifically, for a state vector $\vec{v}$, the transformation $\vec{v}' = U\vec{v}$ satisfies:

$$\|\vec{v}'\| = \|U\vec{v}\| = \|\vec{v}\|,$$

where $\|\cdot\|$ denotes the Euclidean norm. This preservation of norm inherently bounds the propagation of errors and ensures that noise cannot grow arbitrarily during iterative transformations. The preservation of the quantum state's magnitude constrains deviations introduced by errors, effectively limiting their impact. By analyzing rotation-induced discrepancies, particularly azimuthal ($\Delta_{az}$) and elevation ($\Delta_{el}$) angular errors, which emerge during qubit transformations. These errors are of interest as they can counterbalance the accumulation of FHE-induced errors. When the rotation matrix $S(\phi, \theta, \psi)$ is applied iteratively to a state vector $\vec{v}$, it introduces these angular deviations with a predictable periodicity. This behavior is mathematically described as:

$$\Delta_{az}(t) = \Delta_{az}(\vec{v} \cdot SP(t), \vec{v}_{err} \cdot SP(t)),$$

$$\Delta_{el}(t) = \Delta_{el}(\vec{v} \cdot SP(t), \vec{v}_{err} \cdot SP(t)),$$

where $SP(t) = e^{tJ}$ represents the finite rotation matrix generated by $J$, satisfying:

$$J = \begin{bmatrix} 0 & -\phi - \psi & \theta \\ \phi + \psi & 0 & 0 \\ -\theta & 0 & 0 \end{bmatrix}.$$

The eigenvalues of $J$ determine the fundamental frequency of these oscillations, given by:

$$\omega = \frac{2\pi}{\sqrt{\theta^2 + (\phi + \psi)^2}}.$$

In hybrid classical-quantum neural network architectures, the quantum layer plays a role in mitigating noise through periodic error cancellation. This can be understood through the following mechanisms:

- **Quantum Noise Confinement:** Quantum noise, unlike classical noise, evolves under the unitary constraints of $SU(2)$. The preservation of the norm ensures that errors remain bounded, preventing indefinite accumulation. The bounded evolution introduces oscillatory noise patterns, where errors are periodically "reset" to smaller magnitudes.

- **Error Correction Through Hybridization:** The output of the quantum layer, with its stabilized noise, is passed to the classical neural network. The classical network processes the signal while suppressing residual quantum noise, leveraging the strengths of both paradigms to enhance overall error tolerance(Kundu & Ghosh (2024))

The noise propagation retains its periodic or quasi-periodic characteristics for general Euler angles $(\phi, \theta, \psi)$, using infinitesimal rotations $SP(\partial t)$, the time evolution of the system can be expressed as:

$$SP(t) = \exp(tJ), \quad \text{where } J \text{ is traceless and governs periodicity.}$$

This ensures that error propagation does not lead to unbounded growth, instead oscillating with periodicity $\omega$. FHE, which is inherently prone to noise accumulation during encrypted computations, benefits significantly from this framework. By introducing a quantum layer within a hybrid architecture, the controlled periodic evolution of noise prevents uncontrolled error growth. The preservation of the norm in quantum operations further supports this by limiting the magnitude of deviations. This periodic "resetting" mechanism enhances the fidelity of encrypted information, even for deep computations.

## 7.2 EFFECTS OF FHE MODIFICATIONS

In terms of machine learning applications, the key parameters influencing performance include the `bit scale`, `poly modulus degree`, and `number of extremas`, which together determine how many values can be encrypted and processed within a given system.

The main insight from the above table is that reducing parameters like the bit scale, polynomial modulus degree, or extrema count enables encrypting more values but often comes at the expense of security. For example, decreasing the bit scale from 40 to 20 increases the number of encryptable values to $690,000$, but it also lowers the security level. Similarly, a smaller polynomial modulus degree allows for encrypting more values, but it reduces both the security level and the depth of operations. In FHE systems, communication costs are as critical as computational efficiency, especially when encrypted data is transmitted, such as in cloud-based machine learning scenarios.

| Bit Scale | Polymodulus Degree | Extrema Count | Encrypted Parameter Count | Time (s) |
|---|---|---|---|---|
| 40 | 32768 | 8 | 15000 | 290 |
| 30 | 16384 | 8 | 15000 | 17 |
| 20 | 8192 | 8 | 20000 | 319 |
| 20 | 8192 | 1 | 50000 | 209 |
| 30 | 8192 | 1 | 85000 | 338 |
| 40 | 4096 | 1 | 180000 | 440 |
| 40 | 8192 | 1 | 140000 | 326 |
| 30 | 8192 | 1 | 350000 | 391 |
| 20 | 4096 | 1 | 410000 | 339 |
| 20 | 32768 | 1 | 690000 | 432 |

Table 5: Impact of encryption parameters (bit scale, polymodulus degree, extrema count) on the number of encrypted values and computational time, highlighting trade-offs between security, efficiency, and scalability.

Larger ciphertexts, which result from higher encryption parameters or deeper computational models, require greater bandwidth, increasing communication overhead. Thus, optimizing encryption parameters like bit scale or polynomial degree is crucial for balancing security, computational efficiency, and communication costs. For example, reducing these parameters can minimize ciphertext size and improve efficiency while maintaining an acceptable level of security. Applications like secure machine learning rely on managing computational and communication trade-offs to achieve robust performance and maintain high levels of security. Additionally, a higher global scale provides increased precision for more accurate computations but demands more memory. The coefficient modulus size determines the encryption depth, where smaller modulus sizes enable faster encryption but limit the number of encryptable values and reduce security. Balancing these factors is critical for practical FHE implementations.

## 7.3 EFFICACY OF MQMoE APPROACH

The efficacy of the MQMoE approach is demonstrated through its versatile compatibility with diverse datasets and modalities, as shown in Table 6. This adaptability highlights its potential to handle complex multimodal tasks, integrating text, image, audio, and video inputs effectively. A notable example is the RAVDESS (Ryerson Audio-Visual Database of Emotional Speech and Song) dataset, which is widely used in emotion recognition research.

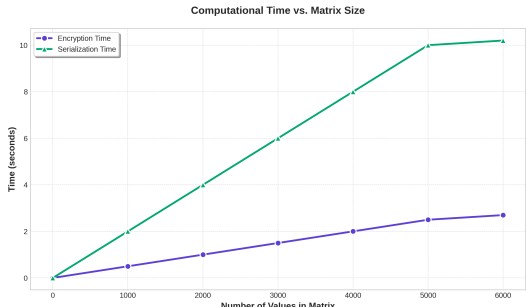

Figure 6: Comparision of matrix size growth and time required for encryption.

The dataset contains audio-visual recordings of 24 professional actors (12 male and 12 female), each expressing eight emotions: calm, happy, sad, angry, fearful, surprised, disgust, and neutral. The data is synchronized, with both speech and song modalities, making it ideal for multimodal emotion recognition tasks that require the joint analysis of audio and visual cues. This richness in data supports the evaluation of the MQMoE approach in handling real-world, emotionally nuanced inputs across different modalities.

| Dataset/Modality | Text | Image | Audio | Video |
|------------------|------|-------|-------|-------|
| **CIFAR-10** | ✕ | ✓ | ✕ | ✕ |
| **DNA Sequence** | ✓ | ✕ | ✕ | ✕ |
| **MRI Scan** | ✕ | ✓ | ✕ | ✕ |
| **PCOS** | ✓ | ✕ | ✕ | ✕ |
| **DNA+MRI** | ✓ | ✓ | ✕ | ✕ |
| **RAVDESS** | ✕ | ✓ | ✓ | ✓ |

Table 6: Framework compatibility with datasets and modalities.

