# OpenReview forum: "MQFL-FHE: Multimodal Quantum Federated Learning Framework with Fully Homomorphic Encryption"
_ICLR.cc/2025/Conference — Submitted to ICLR 2025_

### Official Review · Reviewer_oWfH · 2024-10-31

**Soundness:** 2
**Presentation:** 2
**Contribution:** 1
**Rating:** 3
**Confidence:** 4

**Summary:**

This paper presents an approach to Federated Learning (FL) that incorporates fully homomorphic encryption (FHE) using the Cheon-Kim-Kim-Song (CKKS) encryption scheme, enabling secure parameter aggregation on encrypted data. The authors address a critical challenge in FHE-based federated learning: the introduction of noise during encrypted data aggregation, which affects model accuracy post-decryption. Their key contribution lies in demonstrating how quantum models can effectively mitigate this noise-induced performance degradation.
The authors propose three progressive implementations combining quantum computing with FL+FHE: (i) QFL+FHE: A basic integration of quantum models with FL+FHE, (ii) MQFL+FHE: An extension of QFL+FHE designed to handle multimodal input data, (iii) MQMoE-FL-FHE: An advanced architecture incorporating mixture of experts strategy within the QFL+FHE framework
The experimental validation compare the classical FL+FHE and centralized approaches across multiple datasets, including CIFAR-10, DNA sequences, MRI scans, and PCOS. For multimodal applications, the they evaluate their framework on combined DNA+MRI datasets.

**Strengths:**

The experimental results show that quantum neural networks (QNN) is more robust to be trained in in federated settings. Specifically, when comparing centralized versus federated learning with FHE, classical models show a larger drop in performance than quantum models.

**Weaknesses:**

1- In Algorithm 1, the decryption seems to happen on the server side, which is problematic. If the server has access to the secret key, it can decrypt each client's model parameters individually, making the encrypted aggregation redundant, so the purpose of the encryption should be clarified.
2- The simulation (dashboard_src/client.py) the implementation differs from Algorithm 1 and shows that only clients have access to the secret key. Clients send encrypted model updates to the server, where aggregation occurs on encrypted parameters, and the aggregate value is returned to each client for decryption. However, this scenario has a critical weakness: for the server to aggregate encrypted parameters, all clients must share the same secret key. The paper doesn't specify how clients share this key without server knowledge, particularly if all communication goes through the server. If secret sharing is used, there should be additional communication channels specified; otherwise, the purpose of FHE in this scenario is questionable.
3- The paper lacks theoretical proof demonstrating why quantum models should perform better under FL+FHE compared to classical approaches. Additionally, some simulation results appear inconsistent with theoretical expectations mentioning in point 5.
4- The application of QNN in this work lacks clarity. It appears disconnected from the FL logic and CKKS encryption mechanism, serving only as a comparison between implementing FL+FHE with classical models versus QNN+FL+FHE .
5- Under ideal conditions (perfect communication, infinite rounds), FL can at best match centralized learning's performance. It is unclear why QFL achieves higher accuracy compared to the centralized version (as shown in Tables 3 and 4 for DNA, MRI, PCOS, and multimodal datasets). Also, QFL+FHE outperforms both QFL and centralized quantum approaches in DNA and multimodal cases which is not compatible with FL concept.
6- The communication cost for FHE implementation is not addressed. The paper should explain both the communication and computation complexity of adding FHE to the FL process.

**Questions:**

1- Could you clarify whether the decryption and global model update process occurs on the server side or client side? The algorithm and implementation appear to show different approaches.
2- Does your implementation require clients to share the same secret key? If so, how is this key sharing accomplished securely without server access to this information?
3- If your system employs secret-sharing schemes for key distribution among clients, how do you handle client dropout scenarios? Have you investigated the impact of client unavailability on the secret-sharing mechanism?
4- The paper claims quantum computing helps mitigate FHE noise accumulation. Could you provide a detailed mathematical analysis demonstrating how quantum computing specifically counteracts this noise?
5- There appears to be an inconsistency between test loss and accuracy values in Tables 3 and 4. While accuracy increases, the loss also increases, which seems counterintuitive. Could you explain this apparent contradiction?
6- Your ablation study states that centralized systems outperform federated ones, yet the values in Tables 3 and 4 show the opposite trend. Could you clarify this discrepancy?

---

> ### Author Response · Authors · 2024-12-02
>
> **Comment 1:** *Could you clarify whether the decryption and global model update process occurs on the server side or client side? The algorithm and implementation appear to show different approaches.*
>
> **Response:** The decryption and global model update occur on the client side. After the server performs homomorphic aggregation of the encrypted model updates, it sends the aggregated encrypted model back to all clients. Each client then decrypts the aggregated model using their private key and updates their local model accordingly. This ensures that the server never has access to the decrypted model parameters, preserving the privacy of the clients' data. We revised Algorithm 1 and the corresponding explanation in the manuscript to make this process clear and consistent.
>
> **Comment 2:** *Does your implementation require clients to share the same secret key? If so, how is this key sharing accomplished securely without server access to this information?*
>
> **Response:** Our implementation eliminates the need for clients to share the same private key by utilizing a public-key homomorphic encryption scheme. In this setup, all clients and the server share a common public key for encryption, while each client retains their own private key for decryption, which is never shared. This ensures that the server with access only to the public key, can perform homomorphic operations on encrypted data from multiple clients without being able to decrypt any of it. The key sharing process is securely managed through a trusted setup phase, where public and private keys are generated. The public key is distributed to all participants, while private keys remain securely with their respective clients. Clients encrypt their model updates using the shared public key and transmit the ciphertexts to the server via secure communication channels.
>
> **Comment 3:** *If your system employs secret-sharing schemes for key distribution among clients, how do you handle client dropout scenarios? Have you investigated the impact of client unavailability on the secret-sharing mechanism?*
>
> **Response:** Since clients do not share secret keys in our framework, client dropout does not affect the key distribution or the encryption and decryption processes. Each client operates independently with their own private key.
> If we were to use a secret-sharing scheme for key distribution, client dropout could pose challenges in reconstructing the secret key or maintaining the integrity of the system. However, this is not applicable in our implementation.
>
> **Comment 4:** *The paper claims quantum computing helps mitigate FHE noise accumulation. Could you provide a detailed mathematical analysis demonstrating how quantum computing specifically counteracts this noise?*
>
> **Response:** Thank you for raising this important point. In Appendix Section 7.1, we provide a detailed mathematical analysis demonstrating how quantum computing helps mitigate noise accumulation in FHE-based systems. The analysis highlights the inherent properties of quantum operations governed by the SU(2) group, which enforces unitary constraints that preserve the norm of quantum state vectors. This ensures that errors introduced during computations remain bounded, preventing the unbounded accumulation of noise typically observed in purely classical systems. Moreover, the oscillatory nature of quantum errors, arising from predictable angular deviations during state evolution, introduces a periodic "resetting" mechanism that naturally stabilizes noise growth over time. By incorporating quantum layers within hybrid FHE-quantum architectures, the bounded and oscillatory behavior of quantum noise helps counteract the accumulation of FHE-induced errors. The quantum layers periodically stabilize error propagation, and the subsequent classical processing further suppresses residual quantum noise. This hybrid approach leverages the strengths of both paradigms, with quantum layers ensuring controlled noise dynamics and classical layers refining the output for enhanced fidelity. These mechanisms collectively improve the robustness of encrypted computations, even in deep architectures.

---

> > ### Author Response · Authors · 2024-12-02
> >
> > **Comment 5:** *There appears to be an inconsistency between test loss and accuracy values in Tables 3 and 4. While accuracy increases, the loss also increases, which seems counterintuitive. Could you explain this apparent contradiction?*
> >
> > **Response:** Thank you for pointing this out. While it may seem counterintuitive that the test loss increases alongside an increase in test accuracy, this phenomenon can occur due to the interplay between prediction confidence and correctness in the context of our experiments.
> > In our study, the test loss is calculated using the cross-entropy loss function, which not only penalizes incorrect predictions but also takes into account the confidence level of the model's predictions. Specifically, if the model predicts the correct class with lower confidence (i.e., a lower predicted probability), the loss increases even though the prediction is correct. Conversely, test accuracy simply measures the proportion of correct predictions, without considering the confidence of those predictions.
> > The increase in test accuracy indicates that the model is correctly classifying more instances. However, the simultaneous increase in test loss suggests that the model's confidence in its predictions has decreased. This can happen when the predicted probabilities for the correct classes become less extreme (e.g., decreasing from 0.9 to 0.6). As a result, the cross-entropy loss increases due to the model being less certain, even though it is making more correct predictions.
> > In our experiments, the integration of FHE and QC introduces additional noise and quantization errors into the model parameters and gradients. This noise affects the calibration of the predicted probabilities, causing the model to be less confident in its predictions. The encrypted computations and the approximation errors from simulating quantum operations lead to predicted probabilities that are closer to uniform distributions, increasing the loss.
> > Therefore, the observed increase in test loss alongside an increase in test accuracy is attributed to the model's improved ability to correctly classify samples while exhibiting decreased confidence in those predictions due to the noise introduced by FHE and quantum computations. This explains the apparent contradiction in the results, as the accuracy metric does not account for prediction confidence, whereas the loss function does.
> >
> > **Comment 6:** *Your ablation study states that centralized systems outperform federated ones, yet the values in Tables 3 and 4 show the opposite trend. Could you clarify this discrepancy?*
> >
> > **Response:** Thank you for pointing out the discrepancy between our ablation study and the results in Tables 3 and 4. We have carefully revisited this analysis and clarified the underlying explanation. While centralized systems generally outperform federated systems in terms of computational efficiency and the ease of implementing privacy-preserving techniques, the performance trend can vary depending on the dataset and the quality of the data. Specifically, our revised analysis highlights that federated systems using QML can outperform centralized models in certain scenarios, particularly with smaller datasets. In these cases, QML's ability to generate richer representations allows federated systems to perform better, as it is well-suited for small-scale problems. Additionally, in federated settings with data distributed across clients, QML's ability to handle sparse and fragmented data makes it more effective for decentralized problems, leading to better performance compared to centralized models in these settings. We have updated the manuscript to reflect these insights and ensure consistency with the results presented in the tables.

---

### Official Review · Reviewer_vxVB · 2024-11-02

**Soundness:** 3
**Presentation:** 2
**Contribution:** 2
**Rating:** 5
**Confidence:** 3

**Summary:**

This paper introduces MQFL-FHE, a Multimodal Quantum Federated Learning framework that incorporates Fully Homomorphic Encryption (FHE) to address data privacy concerns while utilizing quantum computing to mitigate performance degradation typically caused by encryption. The proposed framework uniquely integrates a Multimodal Quantum Mixture of Experts (MQMoE) model within the Federated Learning (FL) setup to enhance representational generalizability and classification accuracy, especially for underrepresented categories. Experimental validation is conducted using multimodal datasets, including genomics and brain MRI scans, demonstrating the framework's potential to improve privacy and performance.

**Strengths:**

The paper provides a novel integration of quantum computing and fully homomorphic encryption within the federated learning context, which is an innovative contribution to improving data privacy without sacrificing model performance.
﻿
The proposed MQMoE architecture effectively handles diverse multimodal data, achieving enhanced representation learning and mitigating the performance degradation associated with homomorphic encryption.
﻿
The work is well-supported by experimental results, which illustrate improvements in both data privacy and model accuracy across diverse datasets, particularly in sensitive areas like genomics and MRI scans.

**Weaknesses:**

The paper lacks sufficient detail on the practical implementation challenges associated with deploying the proposed quantum-enhanced federated learning framework.

The paper discusses the impact of homomorphic encryption on model accuracy in the introduction and related works. However, the discussion and citations related to this topic should be expanded to provide a more comprehensive context. Additionally, the experiments could better reflect this aspect by including different hyperparameters or model structures to illustrate the effects more thoroughly.

**Questions:**

The authors demonstrate the framework's performance using DNA and MRI datasets, but the scalability to other types of multimodal data is unclear. Could the authors elaborate on how adaptable the proposed approach is to other types of data and any challenges they foresee?
﻿
The paper discusses the impact of homomorphic encryption on model accuracy. It would be beneficial to expand on this discussion with additional citations to similar works and provide more thorough experiments to investigate the impact. For example, testing different hyperparameters or model structures.

---

> ### Author Response · Authors · 2024-12-02
>
> **Comment:** *The authors demonstrate the framework's performance using DNA and MRI datasets, but the scalability to other types of multimodal data is unclear. Could the authors elaborate on how adaptable the proposed approach is to other types of data and any challenges they foresee? ﻿ The paper discusses the impact of homomorphic encryption on model accuracy. It would be beneficial to expand on this discussion with additional citations to similar works and provide more thorough experiments to investigate the impact. For example, testing different hyperparameters or model structures.*
>
> **Response:** We thank the reviewer for their constructive feedback. Below, we provide clarifications and further details on the adaptability of our framework to other types of multimodal data and the impact of FHE on model accuracy.
> Regarding the scalability of our proposed approach to other types of multimodal data, we agree that this is an important point to address. In Section 7.3 (Efficacy of MQMoE Approach) of the appendix, we demonstrate the versatility of our framework by showing its compatibility with a diverse set of datasets and modalities, including text, image, audio, and video. Specifically, Table 6 illustrates the framework’s ability to handle various data types, such as CIFAR-10 (image), DNA sequences, MRI scans, and the RAVDESS dataset (audio-visual). The RAVDESS dataset, which includes synchronized emotional speech and song recordings, is an excellent example of how the MQMoE approach integrates both audio and visual modalities for emotion recognition. This provides strong evidence of the framework’s capability to manage complex multimodal tasks. In response to the comment on the impact of homomorphic encryption on model accuracy, we have extended our discussion in Section 7.2 (Effects of FHE Modifications). As outlined in Table 5, we provide a detailed analysis of how different encryption parameters (bit scale, polynomial modulus degree, and extrema count) affect both the number of encryptable values and computational efficiency. We highlight that reducing parameters such as the bit scale and polynomial modulus degree increases the number of encryptable values but at the expense of lower security levels. This trade-off is crucial when considering the impact of FHE on model accuracy and efficiency. We acknowledge that further experiments are necessary to better understand these trade-offs, and we plan to extend our work by testing different hyperparameters and model structures to investigate how these modifications influence model performance.

---

### Official Review · Reviewer_ARH6 · 2024-11-04

**Soundness:** 2
**Presentation:** 2
**Contribution:** 2
**Rating:** 3
**Confidence:** 3

**Summary:**

In this paper, the authors propose a multimodal quantum federated learning (FL) framework that incorporates quantum computing to mitigate the performance degradation in the aggregation phase caused by fully homomorphic encryption (FHE). The authors integrate the MQMoE model with FHE in FL to perform task-specific learning. They use the CKKS encryption scheme to encrypt local models and combine it with quantum federated learning to handle heterogeneous data modalities.

**Strengths:**

The authors introduce the research background of quantum computing, privacy protection, and multimodality in federated learning. They propose a framework that combines quantum federated learning and homomorphic encryption and apply it to multimodal quantum federated learning. Through ablation experiments, they demonstrate the specific role of each module.

**Weaknesses:**

1. The paper lacks an in-depth exploration of the integration between homomorphic encryption, quantum computing, and federated learning. There is insufficient discussion on how these technologies work together and how their respective advantages are reflected within the framework.
2. The argumentation regarding homomorphic encryption technology is insufficient. The paper employs the CKKS scheme but lacks a thorough discussion on security analysis and threat models, including potential attack methods and countermeasures, which may weaken the paper's discourse on privacy protection. For example, references such as "Remark on the Security of CKKS Scheme in Practice" by Jung Hee Cheon, Seungwan Hong, and Duhyeong Kim (2020) and "On the Security of Homomorphic Encryption on Approximate Numbers" by Li, B., Micciancio, D. (2021) should be considered.
3. In the experimental section, the paper lacks information on the parameter configuration of the CKKS scheme and its corresponding security levels, which may affect the reproducibility and practicality of the experimental results.
4. Additionally, the paper lacks technical innovation, merely combining homomorphic encryption and quantum computing for use in federated learning.
5. The layout of some figures in the paper could also be improved, as some images are too small to read comfortably.

**Questions:**

1. How do homomorphic encryption, quantum computing, and federated learning work together, and how are their respective advantages reflected within the framework?
2. Without security analysis and threat models, how can one ascertain the risks and limitations of the framework in specific application scenarios?
3. What are the specific security parameters of the CKKS scheme?

---

> ### Author Response · Authors · 2024-12-02
>
> **Comment 1:** *In the experimental section, the paper lacks information on the parameter configuration of the CKKS scheme and its corresponding security levels, which may affect the reproducibility and practicality of the experimental results.*
>
> **Response:** We agree that specifying the parameter configurations of the CKKS scheme is crucial for reproducibility and assessing security levels. In our experiments, we used the following parameters:
>
> * **Polynomial Modulus Degree (n):** 8192. This parameter determines the ring dimension and directly affects both security and computational efficiency. A degree of 8192 offers a balance between performance and security, providing approximately 128 bits of security based on standard lattice-based cryptography estimates.
>
> * **Coefficient Modulus (q):**  A modulus chain composed of four primes with bit-lengths [60, 40, 40, 60], totaling 200 bits. This configuration supports the required multiplicative depth for our computations while maintaining manageable ciphertext sizes.
>
> * **Scaling Factor ($\delta$):** $2^{40}$. The scaling factor is chosen to balance precision and noise growth during homomorphic operations, ensuring that approximate arithmetic remains accurate throughout the computations.
>
> These parameters balance security and computational efficiency, offering approximately 128 bits of security based on standard estimates for lattice-based cryptography.
> Including these details ensures that others can reproduce our experiments and understand the security implications of the parameter choices. We have provided these specifics in the ``Methodology'' section of the manuscript.
>
> **Comment 2:** *The layout of some figures in the paper could also be improved, as some images are too small to read comfortably.*
>
> **Response:** We appreciate your feedback on the figure layouts. We have ensured that all figures are appropriately sized and clearly presented to enhance readability. High-resolution images have been used to ensure that all details are readable.
>
> **Comment 3:** *How do homomorphic encryption, quantum computing, and federated learning work together, and how are their respective advantages reflected within the framework?*
>
> **Response:** In our framework, HE, QC, and FL are integrated to create a secure and efficient system for training machine learning models on decentralized and sensitive data. FL allows multiple clients to collaboratively train a global model without sharing their raw data, thus preserving data privacy and complying with regulations like GDPR. However, the exchanged model updates in FL can still be vulnerable to inference attacks.
> To enhance privacy, we incorporate FHE, specifically the CKKS scheme, which enables computations on encrypted data without decryption. By encrypting the model updates prior to transmission, FHE ensures that the server can aggregate the encrypted updates to form a global model without accessing the underlying plaintext data, which addresses the privacy concerns in FL by protecting model updates from potential eavesdropping or malicious actors.
> However, FHE introduces significant computational overhead due to the complexity of performing operations on encrypted data, which can degrade model performance and increase training times. To mitigate this issue, we integrate QC within our model architecture, which potentially offers computational advantages for certain types of operations and can enhance the expressive power of neural networks through QNNs. By integrating these quantum layers into our model, we aim to counteract the performance degradation caused by FHE.
> In our MQMoE model, QC is employed to process different data modalities, such as MRI images and DNA sequences. Each modality is handled by specialized quantum circuits acting as experts, capturing complex patterns and correlations that might be challenging for classical models. The outputs of these quantum experts are then combined using a gating mechanism, which dynamically weights their contributions based on the input data.
> Our framework's combination of FHE, QC, and FL allows us to maintain strong privacy guarantees without sacrificing model performance. Homomorphic encryption secures the data during transmission and aggregation, FL enables collaborative model training without data centralization, and QC enhances computational efficiency and model expressiveness. The respective benefits of each technology are reflected in the framework through improved data privacy, efficient handling of encrypted computations, and the ability to learn from complex, multimodal datasets in a distributed and secure manner.

---

> > ### Author Response · Authors · 2024-12-02
> >
> > **Comment 4:** *Without security analysis and threat models, how can one ascertain the risks and limitations of the framework in specific application scenarios?*
> >
> > **Response:** While our primary focus is on demonstrating the integration and potential benefits of QC with HE in FL, we acknowledge the importance of security analysis and threat modeling.
> > In our framework, we assume a semi-honest (honest-but-curious) threat model, which is common in FL research. Clients are expected to follow the protocol correctly but may attempt to learn additional information from the data they receive.
> > The CKKS scheme provides security based on the hardness of the RLWE problem under this threat model. However, we recognize that active adversaries or more sophisticated attack vectors require additional considerations, such as incorporating secure multi-party computation techniques or zero-knowledge proofs.
> > While a comprehensive security analysis is beyond the scope of our current work, our framework provides a foundation upon which such analyses can be built, and we encourage future research in this direction.
> >
> > **Comment 5:** *What are the specific security parameters of the CKKS scheme?*
> >
> > **Response:** This question is addressed in our response to comment 1 above.

---

### Official Review · Reviewer_Rf55 · 2024-11-08

**Soundness:** 2
**Presentation:** 2
**Contribution:** 3
**Rating:** 3
**Confidence:** 2

**Summary:**

This paper proposes a multimodal quantum federated learning framework (MQFL-FHE) that utilizes quantum computing to improve the performance of fully homomorphic encryption (FHE) in federated learning (FL). The authors specifically focus on federated learning from multimodal data. The authors show that by using quantum NN layer, the performance degradation of using FHE in federated learning can be alleviated.

The experiments are performed on 4 dataset of different modality and the proposed approach is compared against centralized training, federation training and federated training with FHE.

**Strengths:**

- The paper is easy to follow and I commend the authors for showing results on different types of datasets.
- The results are intriguing
   - Even though quantum layers perform worse than classical training in centralized setting and is only on par in federated setup quantum layers + FHE outperforms federated training in some cases.

**Weaknesses:**

- While the empirical results with QC+FHE+FL are better, the motivation for using QC unclear. What is the main intuition? It is further not clear why QC works in some cases and doesn't work in other cases. Authors should investigate this more.
- The experimental setup is quite vague. It is unclear what is the distribution of and size of the training datasets for each client. One of the key issues in FL is heterogenous setups, that is, different clients may have different distribution and different sizes of training dataset. Authors should evaluate the approach in heterogenous settings.
- The baselines are trivial or ablation of the main method. It is unclear how the proposed method QC+FHE perform, compared to other more advanced methods, say CreamFL + FHE? It would be nice to see some more relevant baselines.

- While paper is easy to follow, certain parts of the paper are unclear or incoherent. See questions.

Overall, I think the experimental results in the paper are nice but the setting is limited. The motivation for the proposed approach is not clear to me. I would like authors to improve on these aspects in future.

**Questions:**

- Is the MQMoE framework designed for paired data, Fig 3? That is does the model take (MRI,DNA) pair as the input in the multimodal experiments?
   - If the dataset is paired, how do you handle missing data? Table 1 has different number of MRI & DNA data, so there may be some missing datasets?
- The test accuracy in multimodal setup is poorer than unimodal case in Table 1? So does this suggest that there is no benefit of having an additional modality?

- In sec 5.1,
  - **Effects of FHE modifications** I can't find experiments corresponding to this ablation? It would be nice to include references to table or figures, if any
  - Similar for **Efficacy of the MQMoE approach**, which experiments should I refer?


- Typo: Line 158-159, what is X? Did you intend to write $g_i(\theta, x_i)$
- What is Q, K, V on line 272? How is it computed?

---

> ### Author Response · Authors · 2024-12-02
>
> **Comment 1:** *The experimental setup is quite vague. It is unclear what is the distribution of and size of the training datasets for each client. One of the key issues in FL is heterogenous setups, that is, different clients may have different distribution and different sizes of training dataset. Authors should evaluate the approach in heterogenous settings.*
>
> **Response:** Thank you for highlighting the need for more clarity in our experimental setup. In our experiments, we distributed the datasets equally among clients to focus on the fundamental effects of integrating QC and FHE in FL. Each client received an equal portion of the training data, ensuring uniform data size and distribution.
> We agree that evaluating the approach in heterogeneous settings is important, as real-world FL scenarios often involve clients with varying data distributions and sizes. In future work, we plan to extend our experiments to include heterogeneous data distributions, such as non-IID data, and clients with different amounts of data to assess the robustness and adaptability of our approach under more realistic conditions.
>
> **Comment 2:** *Is the MQMoE framework designed for paired data, Fig 3? That is does the model take (MRI,DNA) pair as the input in the multimodal experiments? If the dataset is paired, how do you handle missing data? Table 1 has different number of MRI \& DNA data, so there may be some missing datasets?*
>
> **Response:** Yes, our MQMoE framework is designed to handle paired data. In our multimodal experiments, the model takes pairs of MRI images and corresponding DNA sequences as input. Each modality is processed through its specialized expert network within the MQMoE architecture, allowing the model to learn complementary representations from both modalities for the same subject. To address cases where data for one modality may be missing (e.g., an MRI without a corresponding DNA sequence), the framework employs masking strategies during training and inference. Specifically, when a modality is unavailable, its expert network is skipped, and the MQMoE dynamically re-weights contributions from the available modalities, ensuring robust performance. As noted in Table 1, while there are mismatched numbers of MRI and DNA samples due to incomplete datasets, our framework is equipped to handle such scenarios without compromising multimodal integration.
>
> **Comment 3:** *The test accuracy in multimodal setup is poorer than unimodal case in Table 1? So does this suggest that there is no benefit of having an additional modality?*
>
> **Response:** Thank you for your observation. While it may seem that the test accuracy in the multimodal setup is slightly lower than in the unimodal case, the difference is minimal, as highlighted in Table 4. For example, in the QFL + FHE configuration, the test accuracy for the multimodal setup (DNA: 95.31\%, MRI: 87.26\%) is comparable to the unimodal DNA (94.32\%) and MRI (88.25\%) accuracies. These minor variations are expected due to the increased complexity of integrating multiple modalities. However, the multimodal setup provides significant advantages beyond test accuracy.
>
> Specifically, a multimodal pipeline allows for a unified inference process, which is computationally more efficient compared to managing two separate unimodal architectures. Running independent pipelines for DNA and MRI would require duplicating resources and increasing inference time, whereas the multimodal setup consolidates both modalities into a single framework, streamlining the inference process. Additionally, the multimodal approach enhances robustness by leveraging complementary information from both modalities, which can be particularly beneficial in real-world scenarios where a single modality might be insufficient. Thus, the multimodal framework provides both practical and performance-related benefits.

---

> > ### Author Response · Authors · 2024-12-02
> >
> > **Comment 4:** *In sec 5.1, Effects of FHE modifications I can't find experiments corresponding to this ablation? It would be nice to include references to table or figures, if any Similar for Efficacy of the MQMoE approach, which experiments should I refer?*
> >
> > **Response:** Thank you for your keen feedback as we have addressed the concerns by including explicit references to the corresponding experiments in the appendix:
> >
> > **1. Effects of FHE Modifications:** The experiments evaluating the impact of FHE parameter variations (bit scale, polynomial modulus degree, and extrema count) have been detailed in **Appendix Section 7.2**. Table 5 highlights these ablation studies, demonstrating the trade-offs between encryption efficiency, scalability, and security. For example, reducing the bit scale or polynomial modulus degree increases the number of encryptable values but can reduce security and operational depth. These results provide a comprehensive analysis of the parameter tuning required for practical FHE applications.
> >
> > **2. Efficacy of the MQMoE Approach:** We have included detailed experimental evaluations in **Appendix Section 7.3**. Table 6 presents the MQMoE framework's compatibility across various datasets and modalities, underscoring its adaptability to complex multimodal tasks. For example, the RAVDESS dataset is used to showcase the approach’s robustness in handling emotionally nuanced multimodal inputs like synchronized audio and visual data.
> >
> > We hope these additions clarify the experimental evidence supporting our claims. Please let us know if further clarifications are required.
> >
> > **Comment 5:** *Typo: Line 158-159, what is X? Did you intend to write $g_i(\theta,x_{i})$*
> >
> > **Response:** Thank you for the comment. In lines 158-159, $X$ refers to a complex root of unity associated with the cyclotomic polynomial tied to the encryption parameters. The term $g_i(\theta, x_i)$ represents the local update function computed by client $i$, where $\theta$ is the model parameter and $x_i$ is the local data of client $i$.
> >
> > **Comment 6:** *What is Q, K, V on line 272? How is it computed?*
> >
> > **Response:** Thank you for the comment. The multi-head attention mechanism used in MQFL-FHE is based on the work by Vaswani et al. (2017), which introduced the Transformer model. In this mechanism, the query ($Q$), key ($K$), and value ($V$) matrices are computed by applying learned linear projections to the input features $X$. These projections enable the model to attend to different parts of the input sequence and capture dependencies across modalities effectively. Specifically:
> > $Q = XW_Q, \quad K = XW_K, \quad V = XW_V,$ where $W_Q$, $W_K$, and $W_V$ are the projection weight matrices. This allows the model to extract relevant features from diverse input modalities through attention-based mechanisms.

---

### Official Review · Reviewer_8xeU · 2024-11-08

**Soundness:** 2
**Presentation:** 2
**Contribution:** 2
**Rating:** 3
**Confidence:** 2

**Summary:**

This work combines a multimodal quantum mixture of experts (MQMoE) with fully homomorphic encryption (FHE). Integrating FHE with federated learning (FL) is advantageous in data privacy, but results in performance degradation of the aggregated model. This work takes a stab at addressing this issue utilizing quantum computing.

**Strengths:**

The manuscript is well written in general. The integration of quantum federated learning to address performance degradation of FHE scheme is interesting. Experiments, although not detailed enough, seem to support the claim.

**Weaknesses:**

- I do not have any experience with fully homomorphic encryption nor quantum federated learning. However, to me, this work is understood as a proof of concept, with many non-trivial tasks abstracted away. Please refer to the Questions section for more detailed comments.
- Given that all quantum experiments are carried out with Pennylane, it is hard to conclude that the proposed method indeed is beneficial; on the other hand, this work seems to assume client as well as server in FL has access to quantum computer, which seems very farfetched.
- It’s a bit hard to follow without any equation numbers, for instance third point in the questions.

**Questions:**

- Quantum computers are inherently more noisy due to its probabilistic nature. Why would utilizing QC “reduce” noise accumulation in conjunction with FHE? I could not find any intuition or explanation.
- It seems that each client is also assumed to have access to a quantum computer, based on Eq. in line 199. Encoding classical data to a quantum state, especially if the data is multi-modal, sounds quite non-trivial. Can you comment on how this can be achieved?
- Continuing on the above point, multimodal dataset preparation is just “abstracted” in Algorithm 1 with $\mathcal{D}_k \leftarrow \text{PrepareMultimodalDataset(k)}$. But this is highly non-trivial, e.g., how would you encode text and image dataset into a quantum state? I’m quite confused how the local model update for client $k$ is done in line 273 when the local data is already encoded into a quantum state (line 199).
- Is the experimental result in Table 3 without the FHE scheme?
- Given that pennylane is used for quantum experiments, are processes outlined in Figure 3 utilized at all for experiments? If yes, these should be explained in detail. If not, I’m not sure how to interpret the experimental results in conjunction with what is claimed in the main text.

---

> ### Author Response · Authors · 2024-12-02
>
> **Comment 1:** *Quantum computers are inherently more noisy due to its probabilistic nature. Why would utilizing QC “reduce” noise accumulation in conjunction with FHE? I could not find any intuition or explanation.*
>
> **Response:** Thank you for the insightful question. We have addressed this concern by including a detailed explanation in Appendix Section 7.1: Mathematical Analysis of Error Propagation and Noise Stabilization in FHE-Quantum Hybrid Models. In this section, we explain how QC, despite its inherent noise due to the probabilistic nature of qubits, can actually help reduce noise accumulation when combined with FHE. QC processes information using qubits confined to the Bloch sphere, with operations applied through norm-preserving unitary transformations. This limits error propagation in QML when compared to classical ML models, where errors tend to accumulate across computations. QC, when combined with FHE, helps contain the noise introduced by FHE. This is achieved through the periodic reset of error propagation mechanisms in QML. This combination ensures that noise does not significantly affect the accuracy of the computations, leveraging the quantum properties for error containment and FHE for secure, noise-resistant computation.
>
> **Comment 2:** *It seems that each client is also assumed to have access to a quantum computer, based on Eq. in line 199. Encoding classical data to a quantum state, especially if the data is multi-modal, sounds quite non-trivial. Can you comment on how this can be achieved?*
>
> **Response:** Yes, encoding classical, especially multimodal, data into quantum states is non-trivial. In our work, we assume that clients have access to quantum resources, which is increasingly feasible through cloud-based quantum computing platforms (e.g., IBM Quantum Experience, Amazon Braket). For encoding multimodal data, we expanded Section 4.2 of the manuscript to provide a detailed methodology. Specifically, our approach leverages the `PrepareMultimodalDataset` function to preprocess and transform data from distinct modalities into a form suitable for hybrid classical-quantum processing. For example:
>
> * **DNA Sequences:** These are split into *k*-mers to capture sequence patterns and then converted into a textual representation. This textual data is subsequently vectorized using TF-IDF (Term Frequency-Inverse Document Frequency), which provides a sparse numerical representation.
> * **MRI Images:** These are preprocessed using standard transformations such as resizing, normalization, and noise reduction to ensure compatibility across training pipelines.
>
> Once prepared, these datasets are paired to represent multimodal inputs. The next step involves processing the encoded data using a hybrid classical-quantum model. In this model, the classical layers preprocess the data to feed it into the quantum layers, which provide advantages such as higher-dimensional feature representation and enhanced expressivity. This process is facilitated using parameterized quantum circuits (PQC).

---

> > ### Author Response · Authors · 2024-12-02
> >
> > **Comment 3:** *Continuing on the above point, multimodal dataset preparation is just "abstracted" in Algorithm 1 with
> > $D_k \leftarrow \text{PrepareMultimodalDataset}(k).$
> > But this is highly non-trivial. For example, how would you encode text and image datasets into a quantum state? I'm quite confused about how the local model update for client $k$ is done in line 273 when the local data is already encoded into a quantum state (line 199).*
> >
> > **Response:** We recognize that multimodal dataset preparation is a non-trivial process, and we have expanded the explanation in the manuscript to detail how each modality is processed.
> > Specifically, we describe how:
> > Each modality is preprocessed according to its data type (e.g., images are resized and normalized, text is tokenized and vectorized).
> > The processed data is then encoded into quantum states using appropriate encoding schemes.
> > These quantum states are used in the local model update for client k, involving training QNNs capable of handling multimodal data.
> > We provide mathematical formulations and algorithmic steps to clarify this process, ensuring that the methodology is clearly presented.
> >
> > **Comment 4:** *Given that pennylane is used for quantum experiments, are processes outlined in Figure 3 utilized at all for experiments? If yes, these should be explained in detail. If not, I’m not sure how to interpret the experimental results in conjunction with what is claimed in the main text.*
> >
> > **Response:** Yes, the processes shown in Fig. 3 are integral to our experiments and are fully implemented using PennyLane for the quantum components.
> > In our experiments, we implemented the MQMoE model by integrating classical neural networks with quantum neural network layers simulated using PennyLane. Here's how the processes in Fig. 3 are utilized in our experiments:
> >
> > * **Data processing and classical layers:**  We begin by preprocessing the multimodal data—specifically, MRI images and DNA sequences. The MRI images are processed through classical convolutional and pooling layers to extract spatial features, while the DNA sequences are converted into numerical representations and processed through classical linear layers. This initial processing aligns with the left side of Fig. 3.
> >
> > * **Quantum layers with PennyLane:**  The outputs from the classical layers are then fed into quantum layers which are PQCs designed to act as experts for each data modality. We used PennyLane to construct and simulate these quantum layers. PennyLane allows us to define PQC and seamlessly integrate them with classical ML frameworks.
> >
> > * **Gating mechanism and integration:**  The outputs from the quantum experts are combined using a gating network that employs a multi-head attention mechanism. This gating network dynamically weights the contributions of each quantum expert based on the input data, effectively allowing the model to focus on the most relevant features from each modality. This process corresponds to the central part of Fig. 3.
> >
> > * **Model training within FL and FHE:**  The entire MQMoE model parameters, including those of the quantum layers simulated with PennyLane, are encrypted using FHE schemes when transmitted between clients and the central server.
> >
> > * **Experimental Results Reflection:**  The experimental results presented in the paper directly reflect the performance of this MQMoE model. The integration of PennyLane allowed us to simulate quantum computations effectively, and the use of the MQMoE architecture enabled us to handle the multimodal data proficiently. The improvements in model accuracy and performance metrics reported in the results are attributable to this implementation.

---

> > > ### Comment · Reviewer_8xeU · 2024-12-02
> > >
> > > Dear Authors,
> > >
> > > Thank you for your detailed response. I unfortunately still think that there are too much non-trivialities abstracted away (e.g., dataset preparation), and thus I maintain my original score.
> > >
> > > Thank you,
> > > Reviewer 8xeU

---

> > > > ### Author Response · Authors · 2024-12-03
> > > >
> > > > We acknowledge that encoding classical multimodal data into quantum states is a complex task. However, we want to emphasize that the methods we used are based on established techniques in both QC and data preprocessing. Here's a detailed explanation:
> > > >
> > > > **DNA Sequences (Text Data):**
> > > > * **K-mer Transformation:** We convert DNA sequences into numerical representations using k-mer decomposition, a standard method in bioinformatics. Each DNA sequence is broken down into overlapping subsequences of length k, capturing local sequence patterns.
> > > > * **Vectorization:** The k-mers are then vectorized using one-hot encoding or term frequency-inverse document frequency (TF-IDF), which transforms textual data into numerical feature vectors.
> > > > * **Classical Feature Extraction:** The numerical TF-IDF vectors undergo further processing through a set of classical linear layers. These layers serve as additional feature extractors, transforming the high-dimensional and sparse TF-IDF vectors into a more compact and informative representation. This step helps in capturing complex patterns and reducing dimensionality before quantum encoding.
> > > > * **Quantum Encoding:** The numerical vectors are encoded into quantum states using angle encoding, where each feature value is mapped to the rotation angle of a quantum gate, such as the Pauli-Y rotation, acting on a qubit.
> > > >
> > > > **MRI Scans (Image Data):**
> > > > * **Preprocessing:** MRI images undergo standard preprocessing steps, including normalization, resizing, and augmentation to enhance the dataset and reduce overfitting.
> > > > * **Classical Feature Extraction:** We use CNNs to extract high-level features from the images. These features capture essential spatial and structural information.
> > > > * **Quantum Encoding:** The extracted features are then encoded into quantum states. Similar to the DNA data, we use angle encoding where each feature influences the rotation of qubits in the quantum circuit.

---

> ### Author Response · Authors · 2024-12-03
>
> We recognize that some of these processes are complex. However, each step follows well-documented and validated methodologies in existing literature [1-7]. Our contributions lie in integrating these components into a unified framework to tackle the specific challenge of performance degradation due to FHE in FL.
>
> - [1] Schuld, Maria, Ryan Sweke, and Johannes Jakob Meyer. "Effect of data encoding on the expressive power of variational quantum-machine-learning models." Physical Review A 103.3 (2021): 032430.
>
> - [2] Huang, Hsin-Yuan, et al. "Power of data in quantum machine learning." Nature communications 12.1 (2021): 2631.
>
> - [3] Ovalle-Magallanes, Emmanuel, et al. "Quantum angle encoding with learnable rotation applied to quantum–classical convolutional neural networks." Applied Soft Computing 141 (2023): 110307.
>
> - [4] LaRose, Ryan, and Brian Coyle. "Robust data encodings for quantum classifiers." Physical Review A 102.3 (2020): 032420.
>
> - [5] Innan, Nouhaila, and Muhammad Al-Zafar Khan. "Classical-to-Quantum Sequence Encoding in Genomics." arXiv preprint arXiv:2304.10786 (2023).
>
> - [6] Wang, Aijuan, et al. "HQNet: A hybrid quantum network for multi-class MRI brain classification via quantum computing." Expert Systems with Applications 261 (2025): 125537.
>
> - [7] Senokosov, Arsenii, et al. "Quantum machine learning for image classification." Machine Learning: Science and Technology 5.1 (2024): 015040.

---

### Meta-Review · Area_Chair_J751 · 2024-12-22

**Metareview:**

This work combines a multimodal quantum mixture of experts (MQMoE) with fully homomorphic encryption (FHE). Integrating FHE with federated learning (FL) is advantageous in data privacy, but results in performance degradation of the aggregated model. This work takes a stab at addressing this issue utilizing quantum computing.

The paper lacks sufficient detail on the practical implementation challenges associated with deploying the proposed quantum-enhanced federated learning framework. The paper discusses the impact of homomorphic encryption on model accuracy in the introduction and related works. However, the discussion and citations related to this topic should be expanded to provide a more comprehensive context. Additionally, the experiments could better reflect this aspect by including different hyperparameters or model structures to illustrate the effects more thoroughly.

Hence, I vote for rejection. With all these reviewers' suggestions incorporated, I have no doubt this will be a good work in the future.

**Additional Comments On Reviewer Discussion:**

The authors diligently replied to all reviewers' comments. However, reviewers with high confidence think this paper needs more work to improve.

---

### Decision · Program_Chairs · 2025-01-22

Reject